# Sentinel Lymph Node Gene Expression Signature Predicts Recurrence-Free Survival in Cutaneous Melanoma

**DOI:** 10.3390/cancers14204973

**Published:** 2022-10-11

**Authors:** Lilit Karapetyan, William Gooding, Aofei Li, Xi Yang, Andrew Knight, Hassan M. Abushukair, Danielle Vargas De Stefano, Cindy Sander, Arivarasan Karunamurthy, Monica Panelli, Walter J. Storkus, Ahmad A. Tarhini, John M. Kirkwood

**Affiliations:** 1Department of Cutaneous Oncology, H. Lee Moffitt Cancer Center and Research Institute, Tampa, FL 33612, USA; 2Hillman Cancer Center, Biostatistics Facility, Pittsburgh, PA 15213, USA; 3Department of Pathology, University of Pittsburgh Medical Center, Pittsburgh, PA 15213, USA; 4Department of Medicine, Brigham and Women’s Hospital and Dana Farber Cancer Institute, Boston, MA 02215, USA; 5Department of Medicine, Division of General Internal Medicine, University of Pittsburgh Medical Center, Pittsburgh, PA 15213, USA; 6Faculty of Medicine, Jordan University of Science and Technology, Irbid 22110, Jordan; 7Department of Pathology, Division of Pediatric Pathology, UPMC Children’s Hospital of Pittsburgh, Pittsburgh, PA 15224, USA; 8UPMC Hillman Cancer Center, Pittsburgh, PA 15213, USA; 9Departments of Dermatology and Pathology, Divisions of Dermatopathology and Molecular Genetic Pathology, University of Pittsburgh Medical Center, Pittsburgh, PA 15213, USA; 10Departments of Dermatology, Immunology, Pathology and Bioengineering, University of Pittsburgh Medical Center, Pittsburgh, PA 15213, USA; 11Departments of Cutaneous Oncology and Immunology, H. Lee Moffitt Cancer Center and Research Institute, Tampa, FL 33612, USA; 12Department of Medicine, Division of Hematology/Oncology; University of Pittsburgh Medical Center, Hillman Cancer Center, Pittsburgh, PA 15213, USA

**Keywords:** melanoma, sentinel lymph node, gene expression profile, recurrence free survival

## Abstract

**Simple Summary:**

Despite a negative SLN status, patients with Stage IIB/IIC melanoma demonstrate poor melanoma-specific survival when compared to Stage IIIA patients. In this study, we hypothesized that a molecular profile at the level of the SLN would provide potential insights into risk of disease recurrence and serve as a prognostic biomarker of patient outcomes regardless of the presence of SLN melanoma metastases. Gene expression profiling was performed on SLN biopsies using U133A 2.0 Affymetrix gene chips. Our study suggests a novel 12-gene SLN signature risk score which may predict disease recurrence in cutaneous melanoma patients managed with wide excision of the primary tumor and SLN biopsy.

**Abstract:**

We sought to develop a sentinel lymph node gene expression signature score predictive of disease recurrence in patients with cutaneous melanoma. Gene expression profiling was performed on SLN biopsies using U133A 2.0 Affymetrix gene chips. The top 25 genes associated with recurrence-free survival (RFS) were selected and a penalized regression function was used to select 12 genes with a non-zero coefficient. A proportional hazards regression model was used to evaluate the association between clinical covariates, gene signature score, and RFS. Among the 45 patients evaluated, 23 (51%) had a positive SLN. Twenty-one (46.7%) patients developed disease recurrence. For the top 25 differentially expressed genes (DEG), 12 non-zero penalized coefficients were estimated (CLGN, C1QTNF3, ADORA3, ARHGAP8, DCTN1, ASPSCR1, CHRFAM7A, ZNF223, PDE6G, CXCL3, HEXIM1, HLA-DRB). This 12-gene signature score was significantly associated with RFS (*p* < 0.0001) and produced a bootstrap C index of 0.888. In univariate analysis, Breslow thickness, presence of primary tumor ulceration, SLN positivity were each significantly associated with RFS. After simultaneously adjusting for these prognostic factors in relation to the gene signature, the 12-gene score remained a significant independent predictor for RFS (*p* < 0.0001). This SLN 12-gene signature risk score is associated with melanoma recurrence regardless of SLN status and may be used as a prognostic factor for RFS.

## 1. Introduction

The sentinel lymph node (SLN) represents the most frequent initial site of metastasis for primary cutaneous melanoma [1]. Primary melanoma may induce local immunosuppressive effects in the tumor microenvironment of the SLN, mitigating the host anti-tumor response, and resulting in tumor growth both locally and at distant sites of metastasis [2,3]. Advances in sentinel lymph node evaluation techniques have improved staging of patients with early forms of melanoma, eliminating the need for elective lymphadenectomy as a diagnostic/therapeutic procedure in the setting of occult metastatic disease. Indeed, the Multicenter Selective Lymphadenectomy Trial-II (MSLT-II) showed no overall survival benefit for complete dissection after SLN biopsy [4,5]. 

Various patient- and melanoma-related characteristics affect the incidence of SLN positivity. The risk of lymph node involvement is less than 5% for patients with a tumor thickness of less than 0.8 mm (T1a), rising to 50% for deeper ulcerated >4 mm (T4b) melanomas. The incidence of SLN metastasis decreases with increasing age [6]. Age-related increases in lymphatic permeability due to stromal changes associated with hyaluronan and proteoglycan link protein 1 loss may increase false-negative SLN biopsy rates [7]. The complex lymphatic drainage pattern and technical difficulties of SLN biopsy (SLNB) for melanoma in the Head and Neck (H&N) region also result in a higher incidence of false-negative SLN findings when compared to melanomas located on the trunk and extremities [8,9].

SLN status is an important prognostic factor for melanoma progression. The prognostic value of SLN involvement in melanoma patients has been shown in the Multicenter Selective Lymphadenectomy Trial-I (MSLT-I), which demonstrated five-year survival rates of 72% and 90% in patients with tumor-positive vs. -negative SLNs, respectively [4]. Melanoma involvement of SLNs has been correlated with a decreased paracortical area and the presence of paracortical dendritic cells when compared to non-sentinel LNs [10]. The total number of CD8^+^ T cells in tumor-involved SLNs appears to decrease vs. tumor-free nodes, and CD8^+^ T cells within the melanoma-involved SLN demonstrated increased expression of exhaustion markers, including programmed death ligand-1 (PD-L1) [11,12]. The beneficial effects of systemic immunotherapies and targeted therapies in patients with metastatic disease have led to the pursuit of anti-cytotoxic-T-lymphocyte-associated antigen-4 (CTLA-4), anti-programmed cell death-1 (PD-1) monoclonal antibodies, and BRAF/MEK inhibitors in the adjuvant setting, where these agents have become the standard of care for treating stage III melanoma patients post-surgery, resulting in significant improvements in recurrence-free survival (RFS) [13,14,15]. More recently, adjuvant pembrolizumab has resulted in a 35% reduction in the risk of melanoma recurrence amongst patients with resected Stage IIB/IIC disease and leading to regulatory approval of anti-PD1 in this disease indication [16].

Despite negative SLN status, patients with Stage IIB/IIC melanoma demonstrate poor melanoma-specific survival when compared to Stage IIIA patients. This has prompted the evaluation of other prognostic factors related to the SLN for their relationship with melanoma recurrence or progression. In this study, we developed an SLN gene expression signature score predictive of disease recurrence in patients with cutaneous melanoma. We hypothesized that a molecular profile at the level of the SLN would provide potential insights into risk of disease recurrence and serve as a prognostic biomarker of patient outcomes regardless of the presence of SLN melanoma metastases as evaluated by hematoxylin and eosin (H&E) and immunohistochemistry (IHC). 

## 2. Material and Methods

### 2.1. Data Source and Study Population

Cases were retrieved from the University of Pittsburgh protocol 07-133 “Sentinel Node and Non-Sentinel Lymph Nodes (SLN and non-SLN) Procurement from Melanoma Subjects for Molecular Profiling”. The study was approved by the University of Pittsburgh Institutional Review Board. Patients provided written informed consent for enrollment into this study. Eligible patients were ≥12 years old, had cutaneous melanoma with Breslow thickness of ≥2 mm with or without ulceration, and underwent SLN biopsy as part of the staging procedure per standard of care medical practice. One SLN at least 8 mm in longest dimension was collected and processed for research purposes. The SLN sections were immediately placed in 700 μL of RNA lysis buffer (Qiagen RNeasy kit) and frozen at −80 °C for future mRNA extraction for gene expression profiling. SLN tissue procurement procedure and mRNA expression assays were described previously [17]. Primary melanoma characteristics including location, histology, Breslow thickness, presence/absence of ulceration, number of mitoses, and the degree of tumor-infiltrating lymphocytes (TIL) was retrieved from pathology reports. SLN status (positivity/negativity) was evaluated by H&E and IHC (MART-1/melan-A, gp100, tyrosinase, S100), and the number of positive SLNs and presence of extracapsular extension were recorded. Mutational status of the BRAF, NRAS, NF genes were reported for patients with available data. Information pertaining to follow-up visits, presence/absence of recurrence, and death were obtained from the University of Pittsburgh Melanoma and Skin Cancer Program (MSCP SPORE) database. 

### 2.2. Data Analysis

Gene expression profiling was performed using Affymetrix GeneChip^TM^ Human Genome U133 Plus 2.0 Array consisting of 22,277 probe sets. The following criteria were used for gene analysis: (1) removal of probe sets that did not map to a gene symbol, (2) elimination of genes with restricted ranges and low variance, (3) scaling of distributions to have a mean of 0 and a variance of 1, and (4) averaging multiple probe sets mapping to the same gene, resulting int the identification of 9880 genes of potential interest.

The primary objective of this study was to develop an independent SLN gene expression signature score predictive of disease progression in patients with cutaneous melanoma who had completed SLNB. The primary endpoint was recurrence-free survival (RFS), defined as the time from SLNB to development of disease recurrence or death (whichever occurs first). The secondary endpoint was overall survival (OS), defined as the time from SLNB to death from any cause. SLN gene expression profile was an independent variable, and RFS and OS were dependent variables. The proportional hazards regression coefficients for all 9880 genes were estimated. The top 25 genes were selected by majority vote from 300 bootstrap samples based on the lowest *p* value. Penalized regression coefficients were estimated by leave-one-out cross validation to minimize the partial likelihood [18]. Amongst the top 25 genes identified, penalized regression shrank regression coefficients to zero for 13 of these genes, leaving 12 genes with non-zero coefficients, which were selected for further analysis. The signature was re-estimated for each 10-fold subset and refit to the original data to estimate the optimism in overfitting to the same data. This was the third step in a 3-step cross-validation process. (1) Three hundred bootstrap samples of all 9880 genes were used to sum the number of times each gene made the top 25 list. Twenty-five genes were selected from all 300 bootstrap samples by majority vote. From the final list of the top 25 genes, leave-one-out cross-validation was used to select the non-zero penalized regression coefficients that minimized the partial likelihood. (3) the 12 gene signature was refit to subsets of the original data and the refitted model evaluated in the 10% of cases held out. Appendix A shows 100 most selected genes from each of 300 bootstrap samples where we highlight the times each gene was selected.

The risk score was then calculated from these 12 genes as a linear predictor for RFS. Receiver Operating Characteristic (ROC) curves were used to analyze the effect of prediction score on survival. Harrell’s C-index was estimated as an approximation to the area under the ROC curve adapted for time-to-event data. The C- index was re-estimated after 10-fold cross-validation. A proportional hazards regression model was used to evaluate the association between clinical covariates, gene signature score, and RFS. The data were analyzed with the R Project for Statistical Computing package version 3.6.2. The summary of methods and procedures is illustrated in Figure 1. 

### 2.3. Prognostic Role of 12-Gene Signature Score Using TCGA SKCM Cohort

The prognostic performance of 12-gene signature was assessed using The Cancer Genomic Atlas (TCGA) Skin Cutaneous Melanoma (SKCM) cohort (n = 448) which includes transcriptomics data from the primary/metastatic tumor and regional lymph nodes (RLN) tissue. The final used sample size was 426 patients after excluding patients without raw RNA-seq data (n = 5) and duplicate samples for the same patient (n = 2) as well as patients without survival data (n = 15). To address for the potential variation based on the tissue source and because 12-gene signature score was derived from lymph node, we performed a separate analysis as well using only patients with RLN tissue samples (n = 214). Clinicopathologic and RNA-seq data was retrieved for the TCGA-SKCM cohort from the cBioportal database (http://cbioportal.org/) (accessed on 21 August 2022). The constructed formula from our cohort was used to stratify TCGA melanoma patients into high- and low-risk groups using the median risk score as a cut-off point. 

### 2.4. Functional Enrichment Analysis

To further understand the underlying functions of the top 100 significant genes related to RFS (lowest P value for their hazards regression coefficients) we conducted gene ontology (GO) enrichment analyses. GO analysis identifies involved biological processes (BP), cellular components (CC) and molecular functions (MF) in a certain gene set. Furthermore, the Kyoto Encyclopedia of Genes and Genomes (KEGG) pathway analysis was conducted to identify enriched pathways. Enrichment analyses were done using the DAVID bioinformatics web-based tool [19].

## 3. Results

### 3.1. Patients

In total, 45 patients with cutaneous melanoma were evaluated in this study. Baseline characteristics of patients and their disease are presented in Table 1. The median age (range) at diagnosis was 56 (16–81) years, and 18 (40%) patients were female. The median Breslow thickness (inter-quartile range [IQR]) was 4.1 (3.3–6.0) mm, with ulceration present in 29 (67%) cases, and 20 (44%) of the evaluated specimens characterized with non-brisk TIL. The majority of melanomas (40%) were located in the trunk, and 25 (56%) had nodular histology. Twenty-three (51%) of patients had a positive SLN. Amongst those, four (17%) had 2 SLNs involved by melanoma, and seven (30%) had extracapsular extension. Twenty-one (46.7%) patients developed disease recurrence. Of the patients who demonstrated disease recurrence, 11 (52.4%) had a negative SLN. The median time to recurrence was 7.9 years (95% CI 3.8 years – not reached). Three patients (2 male, >60 years old) developed early metastatic disease. The common characteristics of these melanomas were location in the head and neck region and a nodular histology with residual melanoma of >4 mm Breslow thickness on definitive wide excision. Seventeen (37.8%) patients died, with the median overall survival of 8.8 years. 

### 3.2. Gene Expression Analysis

Among 9880 genes tested for association with clinical outcomes, 21 genes had *p* < 0.001, and 92 had *p* < 0.005. Appendix A shows the heatmap of the top 25 Differentially Expressed Genes (DEG) and highlights the two distinct clusters of progressors and non-progressors. Among the top 25 DEGs, 12 non-zero penalized coefficients were estimated, including nine DEG associated with decreased risk [CLGN (Calmegin), C1QTNF3 (C1q Furthermore, TNF Related 3), ADORA3 (Adenosine A₃ Receptor), ARHGAP8 (Rho GTPase Activating Protein 8), CHRFAM7A (CHRNA7 (Exons 5–10) and FAM7A (Exons A–E) Fusion), ZNF223 (Zinc Finger Protein 223), PDE6G (Phosphodiesterase 6G), CXCL3 (C-X-C Motif Chemokine Ligand 3), and HLA-DRB (Major Histocompatibility Complex, Class II, DR Beta]; by contrast, three DEG were associated with an increased risk of disease progression [DCTN1 (Dynactin-1), ASPSCR1 (Tether For SLC2A4, UBX Domain Containing), and HEXIM1 (Hexamethylene Bisacetamide Inducible 1)] (Table 2, Appendix A). The 12-gene signature score produced a 10-fold cross-validated C index of 0.813, and when split at the median, provided an index that was significantly associated with RFS (log-rank test *p* < 0.0001) (Figure 2). SLN status did not impact the prognostic strength of this risk score, since after stratifying by SLN status, the gene signature risk score remained highly significant (*p* = 0.0002) (Figure 3). The 12-gene signature score was scaled to have mean 0 and standard deviation 1 and was associated with disease progression (HR = 30.07, 95% CI 8.12–111.24, *p* < 0.0001. In univariate analysis, Breslow thickness (HR = 1.41), presence of ulceration (HR = 3.61), SLN positivity (HR = 2.50), number of positive SLNs (6.01) and acral-lentiginous histology compared to nodular (HR = 4.07) were each significantly associated with diminished RFS (all *p* < 0.05) (Table 3). After individually adjusting for all these factors plus the gene signature, the 12-gene score alone remained a significant predictor for RFS (*p* < 0.0001). We further evaluated the role of the 12-gene risk score in overall survival and showed that gene signature score was also associated with overall survival (HR = 4.14, 95% CI = 2.30–7.47, *p* < 0.0001).

### 3.3. Prognostic Role of 12-Gene Signature Score Using TCGA SKCM Cohort

The 12-gene signature was significantly associated with OS in TCGA patients with RLN tissue samples (OS: HR = 1.49, 95% CI = 1.01–2.20, *p* = 0.045) (Figure 4A). In the total melanoma samples, the 12-gene signature was also significantly associated with OS (HR = 1.36, 95% CI = 1.04–1.79, *p* = 0.026) (Figure 4B). The gene signature risk score remained significantly associated with OS after multivariable cox proportional hazards analyses that included the AJCC tumor stage, age, and sex in patients with RLN tissue samples and the total sample (Figure 5A,B).

### 3.4. Functional Enrichment Analysis

GO analyses showed significant enrichment for the following BP terms: “aging”, “glycolytic process”, “cellular response to lipopolysaccharide”, “cell adhesion”, “response to cAMP” as for MF terms “apoptosis”, “HIF-1 signaling pathway” and “Glycolysis/Gluconeogenesis” were enriched. For KEGG pathway analysis, “CXCR chemokine receptor binding” and “cadherin binding” pathways were identified. Figure 6 and Table 4 include further details on enriched terms and involved genes in each term.

## 4. Discussion

Despite advances in immune checkpoint blockade immunotherapy and BRAF/MEK-targeted therapies for clinical management of melanoma, it remains critically important to investigate prognostic markers that may more reliably allow for the stratification of patients according to their increased risk of disease recurrence or death, both in the primary neoplasm and in the regional sentinel lymph node, when available. These biomarkers may ultimately guide more precise decisions for adjuvant systemic therapy and allow for modulation of treatment duration and dosing (i.e., potential dose de-escalation for low-risk patients to reduce concerns for treatment-associated adverse events). Several disease-specific characteristics play an important role in melanoma survival including, but are limited to, primary Breslow thickness, primary tumor ulceration, and SLN status. Considering the importance of SLN in melanoma metastases and the poor prognosis of SLN-positive patients, the study of other characteristics associated with SLN status may afford prognostic value and are clearly warranted. We previously described a 25-gene signature related to melanoma oncogenesis and immunosuppression that discriminated patients with SLN positive and negative disease. The current analysis evaluated the SLN gene expression profiles in relation to disease recurrence and course at a mature follow-up. We identified a novel 12-gene SLN signature score that appears to be prognostic for melanoma recurrence and is independent of all other primary melanoma clinicopathological features, as well as the SLN status.

This risk score included nine genes (CLGN, C1QTNF3, ADORA3, ARHGAP8, CHRFAM7A, ZNF223, PDE6G, CXCL3, HLA-DRB) associated with decreased risk of disease progression and three (DCTN1, HEXIM1, ASPSCR1) associated with increased risk of progression. While several of these genes were previously reported to be associated with the pathogenesis of various solid tumors, some have not been previously described, and warrant further prospective investigation. 

The biological basis of the potential prognostic roles of these genes is worth considering. Dynactin-1 (DCTN1) is one of the subunits of DCTN complex which binds to dynein and has an essential role in intracellular transport of vesicles and organelles [20]. Low expression of DCTN1 in primary tumors has been reported to be associated with favorable prognosis in patients with cutaneous melanoma [21] and low-grade glioma [22]. DCTN1 has been identified as an anaplastic lymphoma kinase (ALK) fusion partner in lung cancer [23]. DCTN1 mutations have also been implicated in some neurodegenerative diseases [24]. Alveolar soft part sarcoma chromosome region, candidate 1 (ASPSCR1) is a tethering protein that functions in the regulation of the glucose transporter GLUT4 [25]. ASPSCR1 and Transcription factor E3 (TFE3) fusion is observed in tumors including alveolar soft part sarcoma (ASPS) and Xp11-associated renal cell carcinoma [26]. However, no direct evidence of differential expression of ASPSCR1 linked to increase risk of progression has been reported in cancers previously. The A3 adenosine receptor (ADORA3) is a G-protein coupled cell surface receptor expressed on many immune cells as well as cancer cells including melanoma [27,28,29,30]. Adenosine receptors play an important role in inflammation and immune system regulation via the modulation of the NF-κβ and PI3K/AKT signaling pathways, T cell activation, and tumor cell apoptosis driven through NF-κβ-, Wnt- and p53-dependent pathways [31,32]. There has been an operational linkage described between the adenosine/adenosine receptor and hypoxia-driven pathways, with adenosine activating hypoxia-inducible factor 1 through ADORA3 receptors expressed by melanoma cells [33]. ADORA1 Inhibition promotes tumor immune evasion by regulating the ATF3-PD-L1 Axis [34]. Rho GTPase Activating Protein 8 (ARHGAP8) belongs to the GTPase-activating proteins (RHOGAPs) family which regulates the function of adhesion molecules and cell motility. Their potential role in carcinogenesis, tumor metastases, and conditioning of the tumor immune microenvironment has been described in the setting of bladder cancer [35]. The ARHGAP8 gene was found to be overexpressed in colon cancer tissue vs. adjacent normal colon tissue in association with cervical cancer invasiveness [36,37]. Germline mutations in ARHGAP8 have been identified in colorectal and breast cancers [36]. Calmegin (CLGN) is a endoplasmic reticulum chaperone protein highly expressed in testis tissues [38]. CLGN methylation has been associated with risk for prostate cancer and breast cancer [39]. C1q and TNF Related 3 (C1QTNF3/CTRP3) is a member of the adipokine superfamily that participates in the regulation of lipid metabolism. CTRP3 stimulates the proliferation of certain cell types including osteosarcoma and chondroblastoma tumor cell lines in vitro [40,41]. CHRNA7 (cholinergic receptor, nicotinic, alpha polypeptide 7, exons 5–10) and FAM7A (family with sequence similarity 7a, exons A–E) fusion (CHRFAM7A) product is a chimeric gene encoding dupα7 whose encoded protein forms an ion channel [42]. Dupα7 has been suggested to antagonize carcinogenesis in non-small cell lung cancer [43]. Hexamethylene Bisacetamide Inducible 1 (HEXIM1) is an inhibitor of positive transcription elongation factor b (P-TEFb) [44]. HEXIM1 expression is strongly correlated with TGFβ/SMAD expression and has been linked to poor survival outcomes in patients with prostate adenocarcinoma [45]. Zinc Finger Protein 223 (ZNF223) is a zinc-finger protein of unknown function. ZNF223 upregulation has been associated with a subset of ovarian carcinomas [46]. Phosphodiesterase 6G (PDE6G) encodes the gamma subunit of cyclic GMP-phosphodiesterase and plays a role in the pathogenesis of retinitis pigmentosa [47]. However, no clear association of PDE6G with cancer has emerged. C-X-C Motif Chemokine Ligand 3 (CXCL3) is a strong neutrophil chemoattractant and inflammatory protein (also known as MIP2b). Neutrophils have been shown to suppress tumor metastasis [48,49] and inflammatory (C-X-C) chemokines have been shown to recruit effector cells to the tumor site. Upregulation of this chemokine may prove to be a pivotal biomarker in lowering the risk of melanoma progression [50]. LA-DRB (Major Histocompatibility Complex, Class II, DR Beta) plays an important role in antigen presentation and has been shown to represent a predictive biomarker for response to anti-PD1 therapy and the overall survival of patients with melanoma [51,52].

A 21-gene expression profile has previously been validated to serve as a predictor of clinical outcomes for patients with breast cancer [53]. Several studies have evaluated the role of primary tumor gene expression profiles with survival outcomes in melanoma patients [54,55,56,57]. However, a consensus prognostic gene expression signature has not been defined. This may in part reflect the lack of prospectively-designed, randomized clinical trials where the role of primary tumor molecular profiles can be better determined. 

Most gene profiling studies have focused on gene expression patterns in primary tumors, with a single study evaluating transcriptional profiles in the SLN. Farrow et al. investigated the SLN immune gene profiles in relation to disease recurrence in patients with resected melanomas. Using the Nanostring PanCancer Immune Profiling Panel, this study identified a 12-gene score predictive of disease recurrence, highlighting the role of immune checkpoint TIGIT upregulation and CXCL16 downregulation in disease recurrence, thus supporting our hypothesis that SLN GEP may identify risk groups that are presently elusive [58].

A major strength of our study is the very mature follow-up interval for the patients who are now at a median follow-up of 8.3 years for disease-free patients. This is the first study highlighting prognostic significance of molecular profiling of SLN in melanoma patients using not only immune genes. Our study controlled for other important clinical variables including SLN status and showed that gene expression profile is a strong predictor of disease progression regardless of SLN status. This finding may be particularly relevant for patients with Stage IIB/C/IIIA disease since it remains unclear whether these patients should always receive adjuvant therapy. Limitations for our study include the relatively small cohort of patients evaluated which derived from a single referral center. 

While the 12-gene signature risk score was associated with OS in the TCGA cohort, it is important to note that the TCGA cohort does not serve as an external validation cohort for this study because of differences in tissue source (SLN vs. primary tumor/metastatic tumor and RLN), therefore validation of these hypothesis-generating findings in SLN samples will be required in the future. 

## 5. Conclusions

Our study suggests a novel 12-gene SLN signature risk score which may predict disease recurrence in cutaneous melanoma patients managed with wide excision of the primary tumor and SLN biopsy.

## Figures and Tables

**Figure 1 cancers-14-04973-f001:**
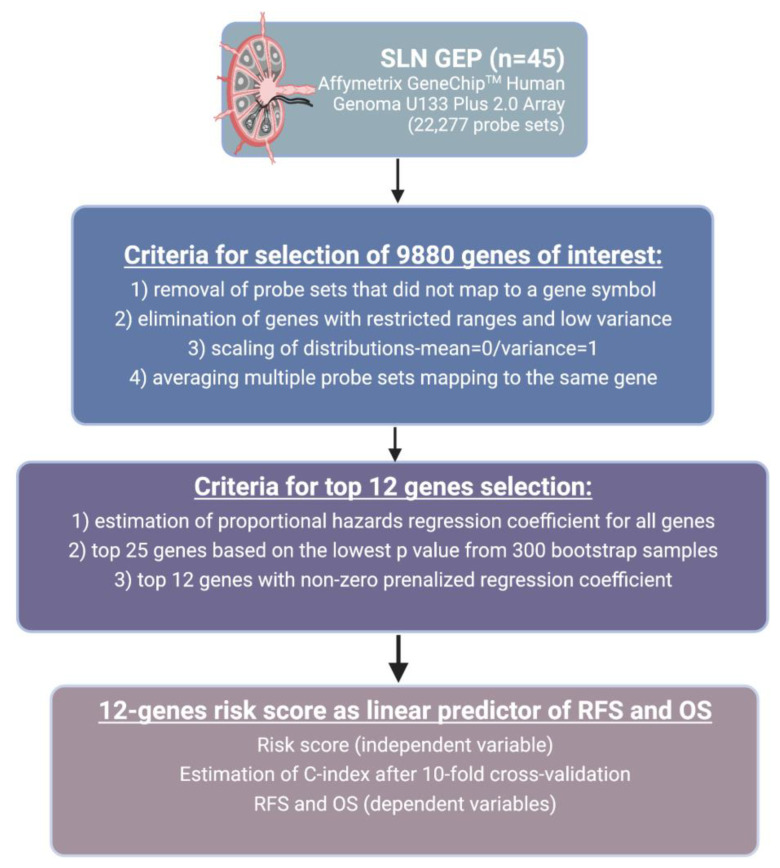
Summary of study methods and procedures. Abbreviations: SLN-sentinel lymph node; RFS-recurrence-free survival; OS-overall survival; GEP-gene expression profiling. Created with BioRender.com.

**Figure 2 cancers-14-04973-f002:**
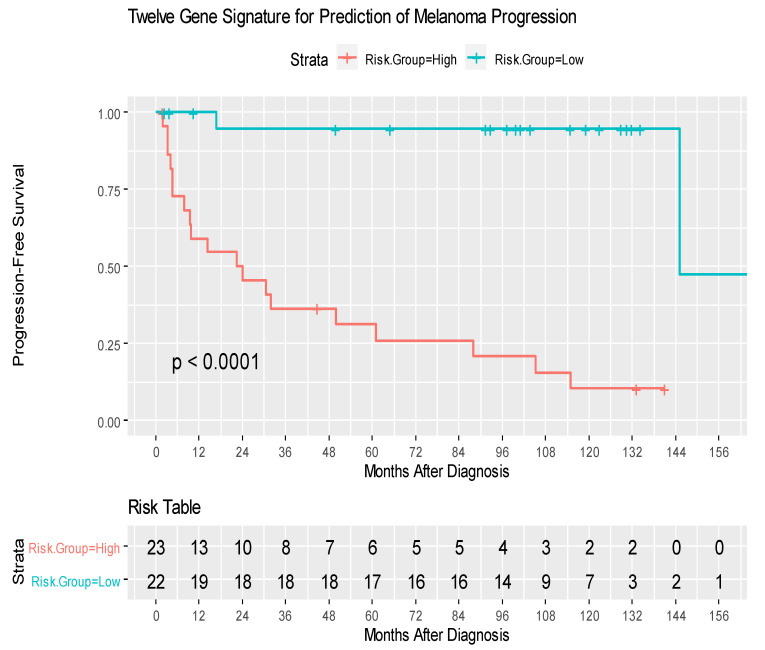
Kaplan–Meier curves of recurrence-free survival amongst patients with High- and Low-risk groups. The risk score was calculated for each patient. The scores from high to low were ranked and split at the median; the ½ of patient with higher scores were labeled the “high risk group”. The high- and low-risk groups were compared, and P-value was calculated by log-rank test.

**Figure 3 cancers-14-04973-f003:**
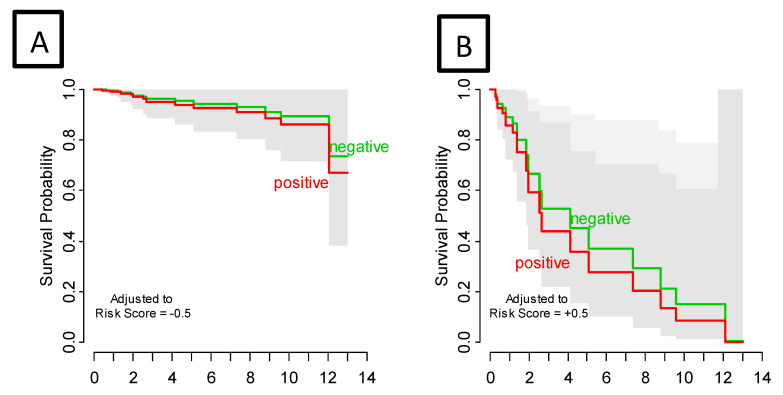
Recurrence-Free Survival by sentinel lymph node status adjusted for gene signature risk score low (**A**) and high (**B**). Survival curves are estimated from a bivariate proportional hazard regression model. Adjusting for risk score, sentinel lymph node status had no effect on PFS (*p* = 0.602); gene signature risk score was highly significant (*p* < 0.0001). Gray bands are (overlapping) 95% confidence intervals.

**Figure 4 cancers-14-04973-f004:**
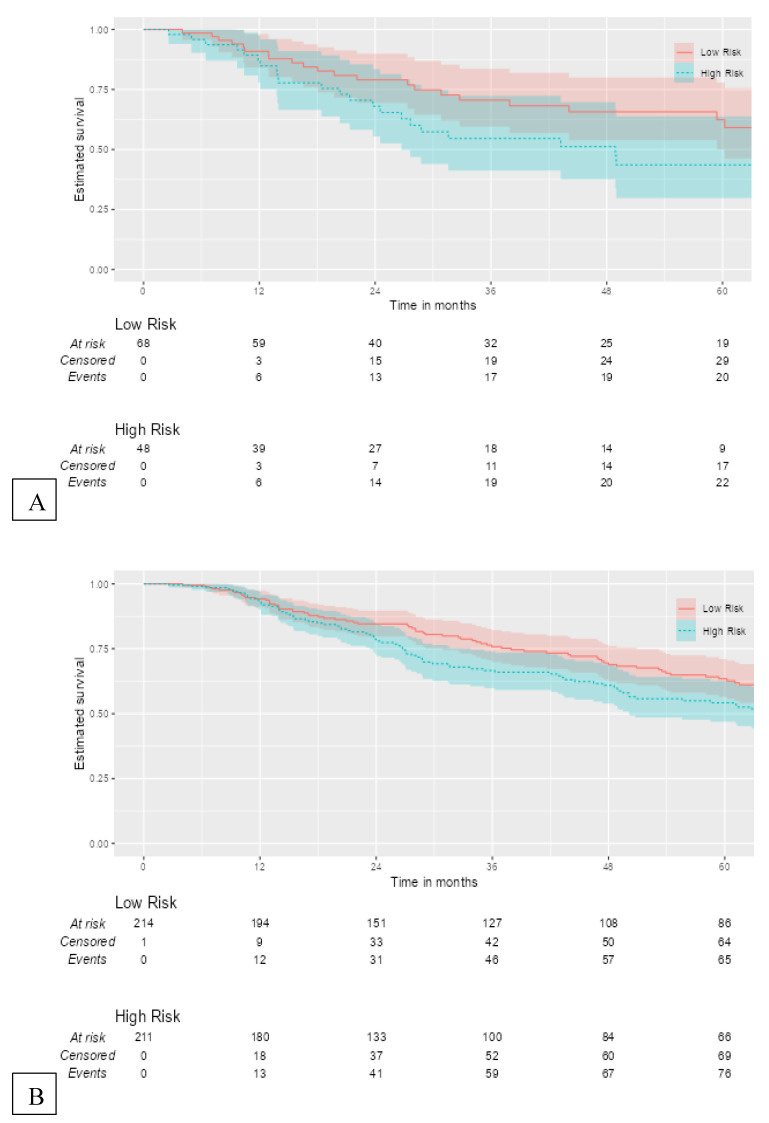
Kaplan–Meier curves for external validation of the 12-gene signature risk score and its survival prognostic performance with (**A**) OS in TCGA melanoma patients with RLN tissue samples (n = 116), and (**B**) OS in the total cohort (n = 426).

**Figure 5 cancers-14-04973-f005:**
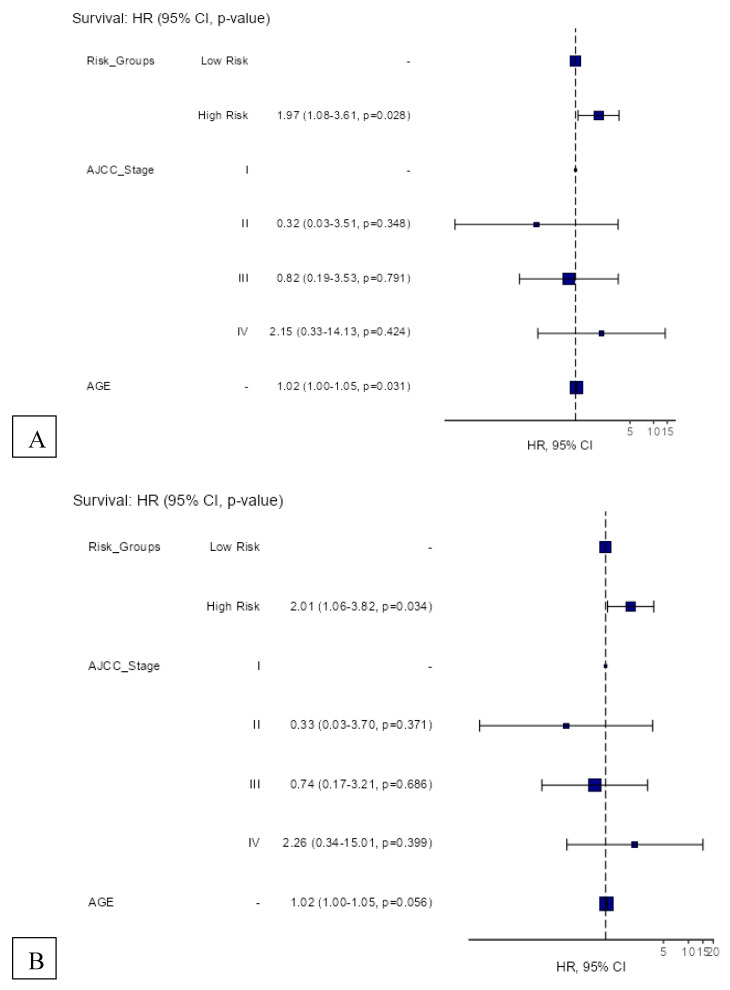
Hazard regression plots for multivariable cox proportional hazards regression analyses of the 12-gene signature risk score and its survival prognostic performance with (**A**) OS in TCGA melanoma patients with RLN tissue samples (n = 116), and (**B**) OS in the total cohort (n = 426).

**Figure 6 cancers-14-04973-f006:**
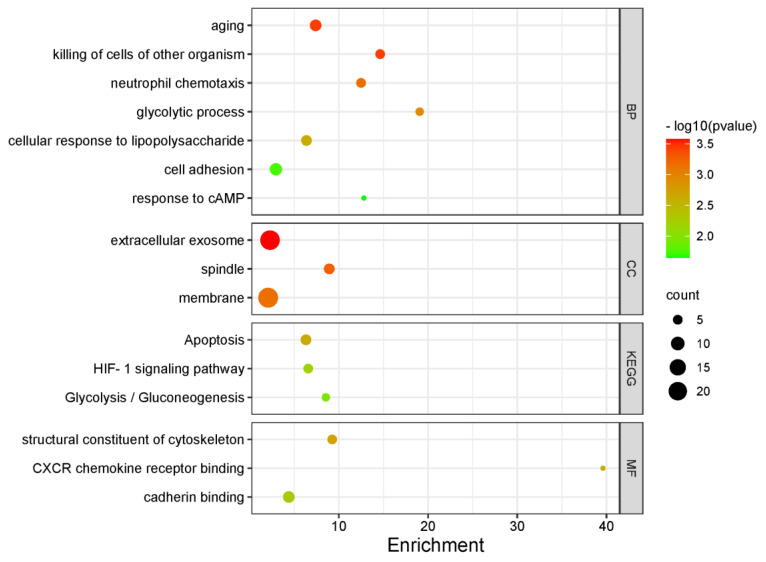
Bubble plot demonstrating significantly enriched GO terms (BP, CC and MF) and KEGG pathways among the top 100 genes with the lowest *p* values for hazards regression coefficients.

**Table 1 cancers-14-04973-t001:** Baseline Characteristics of Patients and Tumors.

Characteristic	Number (%)
Median Age at Diagnosis (range)	56 (16–81)
Gender	
Male	27 (60%)
Female	18 (40%)
Breslow Score median (IQR)	4.1 (3.3–6.0)
Ulceration	
Yes	29 (67%)
No	14 (33%)
TIL	
Absent	7 (16%)
Brisk	5 (11%)
Non-Brisk	20 (44%)
Unknown	13 (29%)
Sentinel Lymph Node Status	
Positive	23 (51%)
Negative	22 (49%)
Number of positive SLN	
1	19 (83%)
2	4 (17%)
Extracapsular Extension	
Yes	7 (30)
No	16 (70)
Location	
Head and Neck	5 (11%)
Trunk	18 (40%)
Upper Extremities	12 (27%)
Lower Extremities	10 (22%)
Histology	
Acral Lentiginous	4 (9%)
Nodular	25 (56%)
Superficial Spreading	8 (18%)
NOS	8 (18%)
Molecular Testing	
BRAF mutant	10(50%)
NRAS mutant	6 (30%)
NF mutant	0 (0)
Triple wild type	4 (20%)

**Table 2 cancers-14-04973-t002:** 12 genes associated with recurrence-free survival with non-zero penalized coefficient (CLGN, C1QTNF3, ADORA3, ARHGAP8, DCTN1, ASPSCR1, CHRFAM7A, ZNF223, PDE6G, CXCL3, HEXIM1, HLA-DRB).

Gene	HR	L95	U95	Penalized Coefficient
CLGN	0.25	0.13	0.50	−0.211
C1QTNF3	0.27	0.14	0.52	−0.236
ADORA3	0.19	0.08	0.45	−0.160
ARHGAP8	0.27	0.14	0.54	−0.067
DCTN1	4.24	1.86	9.68	0.112
ASPSCR1	2.30	1.42	3.74	0.149
CHRFAM7A	0.37	0.20	0.66	−0.112
ZNF223	0.35	0.19	0.66	−0.004
PDE6G	0.21	0.08	0.53	−0.058
CXCL3	0.45	0.27	0.73	−0.096
HEXIM1	2.97	1.45	6.09	0.187
HLA-DRB	0.60	0.42	0.86	−0.124

HR = hazard ratio; L95, (lower) U95 (upper) = 95% confidence bounds; Penalized Coefficient = penalized proportional hazards regression coefficient; A negative coefficient and a hazard ratio < 1 means that high gene expression lowers the risk of progression.

**Table 3 cancers-14-04973-t003:** Proportional hazards regression for the association of progression-free survival with clinical covariates and the gene signature (CLGN, C1QTNF3, ADORA3, ARHGAP8, DCTN1, ASPSCR1, CHRFAM7A, ZNF223, PDE6G, CXCL3, HEXIM1, HLA-DRB).

Covariate	Reference	Hazard Ratio	95 % CI	*p* Value
Gene Signature	−0.72 to + 0.48 *	30.07	8.12–111.24	<0.0001
Breslow thickness	3.3–6.0	1.41	1.14–1.74	0.0014
Number of positive SLNs	0–2	6.01	1.52–23.80	0.0106
Histology	Nodular			0.2549
Acral lentiginous		4.07	1.00–16.47	
NOS		1.24	0.38–4.04	
Superficial spreading		1.02	0.31–3.33	
Ulceration	No	3.61	1.05–12.41	0.0419
SLN status	Negative	2.5	1.01–6.21	0.0485
TIL	Absent			0.3093
Brisk		0.64	0.11–3.86	
Non brisk		1.61	0.45–5.76	
Unknown		0.61	0.14–2.74	
Location	Trunk			0.0764
HN		2.64	0.65–10.62	
LE		2.32	0.74–7.32	
UE		1.38	0.42–4.54	
Sex	Male	1.21	0.49–2.96	0.6817
Age at Diagnosis	49–66	1.93	0.97–3.81	0.0592

* Gene expression was scaled to have mean 0 and standard deviation 1.

**Table 4 cancers-14-04973-t004:** Enriched GO and KEGG pathway terms in the top 100 genes with the lowest *p* values with involved genes in each term.

Class	Terms	Genes	Enrichment	*p* Value
BP	aging	IL10, VCAM1, ELAVL4, TSPO, TIMP1, PAX5, CD68	7.39	0.0004
	killing of cells of other organism	CXCL1, CXCL3, GAPDH, CXCL2, LTF	14.63	0.0004
	neutrophil chemotaxis	CXADR, BSG, CXCL1, CXCL3, CXCL2	12.49	0.0007
	glycolytic process	LDHA, ALDOA, GAPDH, PFKM	19.06	0.0012
	cellular response to lipopolysaccharide	IL10, TSPO, CXCL1, CD68, CXCL3, CXCL2	6.37	0.0024
	cell adhesion	LGALS3BP, COL1A1, CLDN10, VCAM1, SPECC1L, BSG, FOLR2, SIGLEC7	2.94	0.0182
	response to cAMP	COL1A1, LDHA, BSG	12.80	0.0225
CC	extracellular exosome	LGALS3BP, PLVAP, VCAM1, GOT2, HSPB1, LIFR, SNF8, LTBP3, C1QTNF3, LDHA, TUBA1A, TUBB2A, FXYD2, BSG, TSPO, MYH9, TIMP1, ALDOA, GAPDH, EPHB1, HLA-DRB1, EIF3B, LTF	2.28	0.0003
	spindle	DCTN1, SPECC1L, MYH9, HSPB1, KATNB1, AURKC	8.92	0.0005
	membrane	LGALS3BP, GRAMD1B, VCAM1, DCTN1, PRKDC, ELAVL4, ILK, KATNB1, SNF8, PCSK6, C1QTNF3, LDHA, BSG, IL3RA, MLEC, MYH9, SLC39A7, ALDOA, CD68, HRAS, GAPDH, YKT6, PFKM, HLA-DRB1	2.07	0.0007
MF	structural constituent of cytoskeleton	TUBA1B, TUBB2A, TUBA1A, HLA-DRB1, LMNB2	9.25	0.0020
	CXCR chemokine receptor binding	CXCL1, CXCL3, CXCL2	39.60	0.0025
	cadherin binding	GOLGA2, LDHA, BSG, MYH9, ALDOA, YKT6, CDH18	4.39	0.0051
KEGG	Apoptosis	TUBA1B, TUBA1A, CTSK, IL3RA, HRAS, LMNB2	6.31	0.0023
	HIF-1 signaling pathway	LDHA, TIMP1, ALDOA, GAPDH, PFKM	6.56	0.0065
	Glycolysis/Gluconeogenesis	LDHA, ALDOA, GAPDH, PFKM	8.54	0.0108

## Data Availability

The datasets used and/or analyzed during the current study are available from the corresponding authors on a reasonable request.

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
