# Peer review of "Sentinel Lymph Node Gene Expression Signature Predicts Recurrence-Free Survival in Cutaneous Melanoma"

_cancers, 2022, doi:10.3390/cancers14204973_

Round 1

Reviewer 1 Report (Previous Reviewer 1)

I would like to thank the authors for addressing the comments more thoroughly now.

I just had one remaining comment:

During the 10-fold cross validation, the authors use a 12-gene signature. Was the 12-gene signature different for each of the 10 folds, as the training and test data are different for each fold? If done correctly I assume the 12-gene signature set would vary when each fold is used as a test set. The authors should clarify this and also maybe provide a supplementary table on how many times each gene appears as a signature gene. For example, some genes may be a signature gene 10 times, some other gene may be a signature only one time. This can be useful information for biologists.

Author Response

Dear Reviewer,

Re: Resubmission of manuscript cancers- 1960563

Thank you for taking the time to review our manuscript. We appreciate your valuable comment.

We have revised our manuscript, according to the comments and suggestions. Below we provide the response and summarize the updates in the revised manuscript in the highlighted form.

Comment:

During the 10-fold cross validation, the authors use a 12-gene signature. Was the 12-gene signature different for each of the 10 folds, as the training and test data are different for each fold? If done correctly I assume the 12-gene signature set would vary when each fold is used as a test set. The authors should clarify this and also maybe provide a supplementary table on how many times each gene appears as a signature gene. For example, some genes may be a signature gene 10 times, some other gene may be a signature only one time. This can be useful information for biologists.

Response:

The signature was re-estimated for each 10-fold subset and refit to the original data to estimate the optimism in overfitting to the same data. This was the third step in a 3 step cross-validation process. (1) Three hundred bootstrap samples of all 9880 genes were used to sum the number of times each gene made the top 25 list. Twenty-five genes were selected from all 300 bootstrap samples by majority vote.   From the final list of the top 25 genes, leave-one-out cross-validation was used to select the non-zero penalized regression coefficients that minimized the partial likelihood. (3) the 12 gene signature was refit to subsets of the original data and the refitted model was evaluated in the 10% of cases held out. Per the reviewer’s recommendation, the table of the 100 most selected genes from each of 300 bootstrap samples was provided where we highlight the times each gene was selected. The manuscript was updated with the following sentences and highlighted on page 6, lines 13-23 “The signature was re-estimated for each 10-fold subset and refit to the original data to estimate the optimism in overfitting to the same data. This was the third step in a 3-step cross-validation process. (1) Three hundred bootstrap samples of all 9880 genes were used to sum the number of times each gene made the top 25 list. Twenty-five genes were selected from all 300 bootstrap samples by majority vote.   From the final list of the top 25 genes, leave-one-out cross-validation was used to select the non-zero penalized regression coefficients that minimized the partial likelihood. (3) the 12 gene signature was refit to subsets of the original data and the refitted model was evaluated in the 10% of cases held out. Supplementary Table 1 shows 100 most selected genes from each of 300 bootstrap samples where we highlight the times each gene was selected”. The supplementary Table 1 was added to the manuscript on pages 32-34.  

We again want to thank you for all your time and valuable comments.

Sincerely,

Lilit Karapetyan, MD, MS

This manuscript is a resubmission of an earlier submission. The following is a list of the peer review reports and author responses from that submission.

Round 1

Reviewer 1 Report

General comments

In this manuscript, Karapetyan et al. developed a sentinel lymph node gene expression signature score predictive of disease recurrence in patients with cutaneous melanoma. The authors also performed gene expression profiles from sentinel lymph node (SLN) for 45 patients with cutaneous melanoma. 23 out of 45 patients had positive SLN. 21 patients developed disease recurrence.  

The research question asked by the authors is important and interesting. The data generated is valuable, however it is not available for download. However, the bioinformatic analysis performed and the writing requires significant improvements. The validation of the gene signature is weak. I outline my specific comments below:

Major comments

1.     The expression data generated by the authors is valuable.

2.     The writing of the methodology used is very confusing and unclear. Perhaps an overview figure explaining the methodology will help greatly. Also, while describing regression analysis, please let us know what are the dependent and independent variables. While talking about differentially expressed genes (DEGs), please let us know what are the two groups of patients between which differential expression is done.  

3.     How exactly did the authors identify the 25 genes? The first half of the abstract (see sentence ‘The top 25 genes associated with recurrence-free survival (RFS) were selected …’) and methods section seem to suggest that the 25 genes are top genes associated with recurrence free survival. However, the results section and Fig. 1 seem to suggest that these 25 genes are the top DEGs between SLN positive and SLN negative patients.  This is very confusing and needs to be clarified. Also was there a reason why the number ‘25’ was chosen?

4.     If indeed the top 25 genes are DEGs between SLN positive and negative patients (point 3), what is the key insight in using these genes as features to predict recurrence free survival? Is there an assumption that is cancer progressed to SLNs, they are more likely to recur and have a worse prognosis?

5.     The 12 genes which had non-zero coefficients were then used to predict recurrence free survival (RFS) using a linear predictor. This is a supervised approach. I hope my understanding is correct. What was the method used to build a linear predictor? Was potentially confounding factors like age etc. controlled for?

6.     The cross-validation analysis was done using 25/12 genes. However ideally the cross validation should be done by re-identifying the ‘25’ or top genes on the training dataset in each iteration. This will ensure there is no leakage of information from the test data to the training data.

7.     Ideally you would want to test your 12 gene signature on independent datasets. Currently there is no independent validation of the gene signature. This is a significant limitation. If possible, try to predict recurrence free survival using some publicly available datasets. If no such datasets are available, the lack of independent validation should be mentioned as a limitation in the discussion section.

8.     What are the enriched pathways of the 25 genes and 12 genes? Maybe show this as a figure.

9.     The authors have not made the data used in this study available. They say ‘The datasets used and/or analyzed during the current study are available from the corresponding authors on reasonable request’. Given that the most important contribution of the paper is the new dataset they generated, I feel it is important to make the data available (gene expression and phenotypic data) for researchers to use without explicitly asking for it.

Minor comments

a. Figs. 1 and 2 don’t seem to convey much useful information. If space is a problem, they can be moved to the supp. material.

Author Response

In this manuscript, Karapetyan et al. developed a sentinel lymph node gene expression signature score predictive of disease recurrence in patients with cutaneous melanoma. The authors also performed gene expression profiles from sentinel lymph nodes (SLN) for 45 patients with cutaneous melanoma. 23 out of 45 patients had positive SLN. 21 patients developed disease recurrence.  

The research question asked by the authors is important and interesting. The data generated is valuable, however, it is not available for download. However, the bioinformatic analysis performed and the writing requires significant improvements. The validation of the gene signature is weak. I outline my specific comments below:

Major comments

  1. The expression data generated by the authors is valuable.

Response: Thank you for highlighting the importance of our work.

  1. 2.     The writing of the methodology used is very confusing and unclear. Perhaps an overview figure explaining the methodology will help greatly. Also, while describing regression analysis, please let us know what are the dependent and independent variables. While talking about differentially expressed genes (DEGs), please let us know what are the two groups of patients between which differential expression is done.

  Response: The dependent variable for proportional hazards regression was progression-free survival; the independent variable was SLN gene expression. The use of the term differentially expressed genes (DEGs) was misleading and inappropriate.  We apologize for the confusion.

  1. How exactly did the authors identify the 25 genes? The first half of the abstract (see sentence ‘The top 25 genes associated with recurrence-free survival (RFS) were selected …’) and methods section seem to suggest that the 25 genes are top genes associated with recurrence free survival. However, the results section and Fig. 1 seem to suggest that these 25 genes are the top DEGs between SLN positive and SLN negative patients.  This is very confusing and needs to be clarified. Also was there a reason why the number ‘25’ was chosen?

      Response: The selection of the 25 genes is explained in the methods section.  The top 25 genes were selected by majority vote from 300 bootstrap samples based on the lowest p-value.  That is, we listed the 25 genes with the lowest regression p values for each of the 300 bootstrap samples.  Then we counted the number of times each gene was selected over the 300 bootstrap samples, taking the most prevalent 25 genes as our “top 25 list”. We mislabeled Figure 1 by calling these genes “differentially expressed”.  We will revise the title accordingly.

  1. If indeed the top 25 genes are DEGs between SLN positive and negative patients (point 3), what is the key insight in using these genes as features to predict recurrence free survival? Is there an assumption that is cancer progressed to SLNs, they are more likely to recur and have a worse prognosis?

     Response: We did not compare the gene expression in positive vs negative SLNs and the top 25 genes were selected using gene expression of the sentinel lymph node irrespective of whether the SLN was positive or negative.

  1. The 12 genes which had non-zero coefficients were then used to predict recurrence free survival (RFS) using a linear predictor. This is a supervised approach. I hope my understanding is correct. What was the method used to build a linear predictor? Was potentially confounding factors like age etc. controlled for?

     Response:  The method used to build the linear predictor was to construct a linear combination by multiplying the 12 non-zero penalized regression coefficients by the patient’s expression of these 12 genes.   Once the gene signature risk score was established, we added clinical covariates, such as Breslow Score, histology, and SLN positivity to test whether they added value to the gene signature

  1. The cross-validation analysis was done using 25/12 genes. However ideally the cross validation should be done by re-identifying the ‘25’ or top genes on the training dataset in each iteration. This will ensure there is no leakage of information from the test data to the training data.

     Response: The top 25 gene selection process re-identified a new set of 25 genes from each of 300 bootstrap samples.  The area under the ROC curve of the final gene signature was cross-validated by leave-one-out cross-validation. We believe the combination of these two methods is sufficiently robust to avoid overfitting.

  1. Ideally you would want to test your 12 gene signature on independent datasets. Currently there is no independent validation of the gene signature. This is a significant limitation. If possible, try to predict recurrence free survival using some publicly available datasets. If no such datasets are available, the lack of independent validation should be mentioned as a limitation in the discussion section.

      Response: We thank the reviewer for highlighting this point. Indeed, external validation in this scenario is very important. Due to the unavailability of public data containing molecular profiling of both negative and positive SLNs and the limited time provided by the journal for submission of the revised version we will not be able to submit data on the external validation cohort but will perform this during our future work. We assessed the prognostic significance of 12 gene signatures using The Cancer Genomic Atlas (TCGA) melanoma (SKCM) cohort. The methods and results were updated accordingly. While the 12-gene signature risk score was associated with OS in the TCGA cohort, it is important to note that TCGA cohort does not serve as an external validation cohort for this study because of differences in tissue source (SLN vs primary tumor/metastatic tumor and RLN), therefore validation of these hypothesis-generating findings in SLN samples will be required in future. We acknowledge independent SLN (positive and negative SLN) dataset validation as a limitation in the discussion area. 

  1. What are the enriched pathways of the 25 genes and 12 genes? Maybe show this as a figure.

Response: To address this point we conducted gene ontology and KEGG pathway enrichment analyses. However, considering that those enrichment analyses could not retrieve any results with such relatively small gene sets (25/12 genes), we performed the analysis using the top 100 genes. The methods and results were updated accordingly.  

  1. The authors have not made the data used in this study available. They say ‘The datasets used and/or analyzed during the current study are available from the corresponding authors on reasonable request’. Given that the most important contribution of the paper is the new dataset they generated, I feel it is important to make the data available (gene expression and phenotypic data) for researchers to use without explicitly asking for it.

      Response: According to the original IRB submission and institutional policy the data used in this study should be kept confidential and be shared only upon reasonable request and after submission of a request to IRB. We understand the importance of this data contribution and will be glad to share the data in the future on a reasonable request for future collaborations.

Minor comments

  1. 1 and 2 don’t seem to convey much useful information. If space is a problem, they can be moved to the supp. material.

Response: Figures 1 and 2 were moved to the supplementary material.

Reviewer 2 Report

The paper is well written and provides an interesting view. Instead of a gene signature of the primary tumor, now the sentinel node is analyzed. This is an interesting perspective. Also the gene expression signature seems to provide good discrimination between progressors or not. However, as the authors state this is a relatively small cohort and the gene signature is not validated, which is essential to say something about prediction/prognostic purposes (and therefore the effect is probably overestimated). Although the authors state that the findings are mainly 'hypothesis generating', I think for these kind of signatures you need a cohort for generation of the signature and a cohort to validate if the genes do still show some relation with outcome. This should be emphasized in the discussion. And preferably the findings should be validated. 

Furthermore, could the other clarify if a full sentinel node is used for research purposes and how they excluded that these nodes were indeed not positive? 

Ref 56 can be removed. It does not refer to a gene expression profile in cutaneous melanoma. 

Can they elaborate more on why they find such a different genes expressed then the other profile by Farrow et al. 

Author Response

Comments and Suggestions for Authors

  • The paper is well written and provides an interesting view. Instead of a gene signature of the primary tumor, now the sentinel node is analyzed. This is an interesting perspective. Also the gene expression signature seems to provide good discrimination between progressors or not. However, as the authors state this is a relatively small cohort and the gene signature is not validated, which is essential to say something about prediction/prognostic purposes (and therefore the effect is probably overestimated). Although the authors state that the findings are mainly 'hypothesis generating', I think for these kind of signatures you need a cohort for generation of the signature and a cohort to validate if the genes do still show some relation with outcome. This should be emphasized in the discussion. And preferably the findings should be validated. 

      Response: We thank the reviewer for highlighting this point. Indeed, external validation in this scenario is very important. Due to the unavailability of public data containing molecular profiling of both negative and positive SLNs and the limited time provided by the journal for submission of a revised version we will not be able to submit data on the external validation cohort but will perform this during our future work. We assessed the prognostic significance of 12 gene signatures using The Cancer Genomic Atlas (TCGA) melanoma (SKCM) cohort. The methods and results were updated accordingly. While the 12-gene signature risk score was associated with OS in TCGA cohort, it is important to note that TCGA cohort does not serve as an external validation cohort for this study because of differences in tissue source (SLN vs primary tumor/metastatic tumor and RLN), therefore validation of these hypothesis-generating findings in SLN samples will be required in future. We acknowledge independent SLN (positive and negative SLN) dataset validation as a limitation in the discussion area. 

  • Furthermore, could the other clarify if a full sentinel node is used for research purposes and how they excluded that these nodes were indeed not positive? 

Response: Full sentinel lymph node was used for research purposes. SLN status (positivity/negativity) was evaluated by H&E and IHC (MART-1/melan-A, gp100, tyrosinase, S100). 

  • Ref 56 can be removed. It does not refer to a gene expression profile in cutaneous melanoma. 

Response: Ref 56 reports the role of gene expression profiling in other cancers such as breast cancer. It highlights that this strategy is being widely adapted in other cancer types and is being used in clinical settings. If the reviewer believes that the reference and statement should be removed from the manuscript after our explanation, we will be glad to do so.

  • Can they elaborate more on why they find such a different genes expressed then the other profile by Farrow et al. 

Farrow et al. used NanoString Pancancer Immune Profiling Panel which includes only a limited number of immune-related genes (n=730). Our study includes the identification of 9880 genes of potential interest.

Round 2

Reviewer 1 Report

General comments

The authors addressed some of my earlier comments (which I appreciate) but not all of them. However, I was surprised to see that that some of easy-to-address suggestions were not addressed with no explanation on why it was not addressed. Some of these comments were about improve/clarify the writing and it was important. Even for the comments that were addressed, the authors do not tell us (in their reviewer response document) where in the manuscript (like page number etc.) it was addressed and what exactly was the changes made. This is quite surprising as it is so easy to do.  Also, the cross-validation procedure used is still problematic especially given that there is no independent validation.

Specific comments

a.     The authors did not really address some of my comments. For example my previous comment was as follows:

“The writing of the methodology used is very confusing and unclear. Perhaps an overview figure explaining the methodology will help greatly. Also, while describing regression analysis, please let us know what are the dependent and independent variables. While talking about differentially expressed genes (DEGs), please let us know what are the two groups of patients between which differential expression is done”

In the revised draft there is no overview figure (detailing their method and procedure) provided as I suggested. Although the authors now discuss what is the dependent and independent variable in their response, they do not mention whether they have now clarified it in the manuscript. If the authors felt, there is no need to revise their draft then they should at least provide an explanation. This is quite an easy suggestion address, especially important because of unclear writing.         

b.     The authors say “Response: The top 25 gene selection process re-identified a new set of 25 genes from each of 300 bootstrap samples.  The area under the ROC curve of the final gene signature was cross-validated by leave-one-out cross-validation. We believe the combination of these two methods is sufficiently robust to avoid overfitting.”

The procedure to identify the final 25 genes is fine if the validation is done on independent datasets. However for the sake of cross validation, the test data should be kept out and the training (including the bootstrap procedure) needs to be done only on the training data. I do not think this will take much computational time, so no reason to avoid doing this the correct way. This is specifically important given the lack of any independent validation in the paper.

c.     The lack of independent validation remains a significant limitation, although I understand that the authors cannot do much about it. They have acknowledged this in the manuscript which is good.

d.     There was no response to my previous comment on whether the authors controlled for potentially confounding factors like age when they tried to predict recurrence free survival. Do the results hold if such potentially confounding factors are controlled for? Given the lack of independent validation data, it is important to do whatever validation is possible as thoroughly as possible.

Author Response

We appreciate all the valuable comments from the reviewer of our work. We have revised our manuscript, according to the reviewer’s comments, questions, and suggestions. We believe that the manuscript has been further improved. Below we provide the responses and summarized the updates in the revised manuscript in the highlighted form.

Comment:

  1. The authors did not really address some of my comments. For example, my previous comment was as follows: “The writing of the methodology used is very confusing and unclear. Perhaps an overview figure explaining the methodology will help greatly. Also, while describing regression analysis, please let us know what are the dependent and independent variables. While talking about differentially expressed genes (DEGs), please let us know what are the two groups of patients between which differential expression is done” In the revised draft there is no overview figure (detailing their method and procedure) provided as I suggested. Although the authors now discuss what is the dependent and independent variable in their response, they do not mention whether they have now clarified it in the manuscript. If the authors felt, there is no need to revise their draft then they should at least provide an explanation. This is quite an easy suggestion address, especially important because of unclear writing.       

Response:

 Thank you for your suggestion of adding a figure highlighting the main methods and procedures of this manuscript. The Figure 1 was added in the revised draft (page 23 of the main file). The manuscript was revised with the following sentence on the page 6, line 21 “The summary of methods and procedures is illustrated in Figure 1”.  

The clarification regarding dependent and independent variables was added in the revised draft. The manuscript was revised with the following sentence on the page 6, lines 6-7 “SLN gene expression profile was an independent variable, and RFS and OS were dependent variables”.

Comment:

  1. The authors say “Response: The top 25 gene selection process re-identified a new set of 25 genes from each of 300 bootstrap samples.  The area under the ROC curve of the final gene signature was cross-validated by leave-one-out cross-validation. We believe the combination of these two methods is sufficiently robust to avoid overfitting. “The procedure to identify the final 25 genes is fine if the validation is done on independent datasets. However, for the sake of cross validation, the test data should be kept out and the training (including the bootstrap procedure) needs to be done only on the training data. I do not think this will take much computational time, so no reason to avoid doing this the correct way. This is specifically important given the lack of any independent validation in the paper.

Response:

We apologize for not being clearer about the cross-validation.  To insure we are testing the fitted model to a test set that is held out of the training set we used 10-fold cross-validation of the 12 gene model. The original C index was 0.898; after cross-validation this index was reduced to 0.813.  The manuscript was updated with the following sentence in the data analysis section on the page 6, lines 17-18 “The C- index was re-estimated after 10-fold cross-validation”. The Results section was updated with the following sentence on the page 8, lines 17-20 “The 12-gene signature score produced a 10-fold cross-validated C index of 0.813, and when split at the median, provided an index that was significantly associated with RFS (log-rank test p<0.0001).

Comment:

  1. The lack of independent validation remains a significant limitation, although I understand that the authors cannot do much about it. They have acknowledged this in the manuscript which is good.

Response:

We acknowledge the lack of independent validation of our findings. The manuscript was updated with the following sentence on the page 13, lines 3-7 “While the 12-gene signature risk score was associated with OS in TCGA cohort, it is important to note that TCGA cohort does not serve as an external validation cohort for this study because of differences in tissue source (SLN vs primary tumor/metastatic tumor and RLN), therefore validation of these hypothesis-generating findings in SLN samples will be required in future”.

Comment:

  1. There was no response to my previous comment on whether the authors controlled for potentially confounding factors like age when they tried to predict recurrence free survival. Do the results hold if such potentially confounding factors are controlled for? Given the lack of independent validation data, it is important to do whatever validation is possible as thoroughly as possible.

Response:

Once the gene signature risk score was established, we added clinical covariates, such as Breslow thickness, number of positive nodes, ulceration, Mitoses, location, age and gender, histology, and SLN positivity to test whether they added value to the gene signature, controlling for confounding variables. Table 3 (page 21) highlights all potential confounding factors in the main file.  The manuscript was revised with the following sentences highlighted on page 8, lines 24-29 “In univariate analysis, Breslow thickness (HR=1.41), presence of ulceration (HR=3.61), SLN positivity (HR=2.50), number of positive SLNs (6.01) and acral-lentiginous histology compared to nodular (HR=4.07) were each significantly associated with diminished RFS (all p<0.05) (Table 3). After individually adjusting for all these factors plus the gene signature, the 12-gene score alone remained a significant predictor for RFS (p<0.0001)”.

We would like to thank you again for taking the time to review our manuscript.

Reviewer 2 Report

None